# Link between Genotype and Multi-Organ Iron and Complications in Children with Transfusion-Dependent Thalassemia

**DOI:** 10.3390/jpm12030400

**Published:** 2022-03-04

**Authors:** Antonella Meloni, Laura Pistoia, Paolo Ricchi, Maria Caterina Putti, Maria Rita Gamberini, Liana Cuccia, Giuseppe Messina, Francesco Massei, Elena Facchini, Riccardo Righi, Stefania Renne, Giuseppe Peritore, Vincenzo Positano, Filippo Cademartiri

**Affiliations:** 1Department of Radiology, Fondazione G. Monasterio CNR-Regione Toscana, 56124 Pisa, Italy; antonella.meloni@ftgm.it (A.M.); laura.pistoia@ftgm.it (L.P.); positano@ftgm.it (V.P.); 2U.O.C. Bioingegneria, Fondazione G. Monasterio CNR-Regione Toscana, 56124 Pisa, Italy; 3U.O.S.D. Malattie Rare del Globulo Rosso, Azienda Ospedaliera di Rilievo Nazionale “A. Cardarelli”, 80131 Napoli, Italy; pabloricchi@libero.it; 4Dipartimento della Salute della Donna e del Bambino, Clinica di Emato-Oncologia Pediatrica, Azienda Ospedaliero-Università di Padova, 35128 Padova, Italy; mariacaterina.putti@unipd.it; 5Unità Operativa di Day Hospital della Talassemia e delle Emoglobinopatie, Dipartimento della Riproduzione e dell’Accrescimento, Azienda Ospedaliero-Universitaria “S. Anna”, 44124 Cona, Italy; m.gamberini@ospfe.it; 6U.O.C. Ematologia con Talassemia, ARNAS Civico “Benfratelli-Di Cristina”, 90134 Palermo, Italy; liana.cuccia@arnascivico.it; 7Centro Microcitemie, Grande Ospedale Metropolitano “Bianchi-Melacrino-Morelli”, 89100 Reggio Calabria, Italy; gspmessina@virgilio.it; 8Unità Operativa Oncoematologia Pediatrica, Azienda Ospedaliero Universitaria Pisana, 56126 Pisa, Italy; f.massei@med.unipi.it; 9Unità Operativa di Pediatria—Prof A Pession Programma di Oncologia, Ematologia e Trapianto Azienda Ospedaliero-Universitaria di Bologna—Policlinico “S. Orsola-Malpighi”, 40138 Bologna, Italy; elena.facchini@aosp.bo.it; 10Diagnostica per Immagini e Radiologia Interventistica, Ospedale del Delta, 44023 Lagosanto, Italy; riccardo.righi@ausl.fe.it; 11Struttura Complessa di Cardiologia-UTIC, Presidio Ospedaliero “Giovanni Paolo II”, 88046 Lamezia Terme, Italy; stefania.renne@virgilio.it; 12U.O.C. di Radiologia, ARNAS Civico “Benfratelli-Di Cristina”, 90134 Palermo, Italy; giuseppe.peritore@hotmail.it

**Keywords:** transfusion-dependent thalassemia, genotype, iron overload, complications

## Abstract

We evaluated the impact of the genotype on hepatic, pancreatic and myocardial iron content, and on hepatic, cardiac and endocrine complications in children with transfusion-dependent β-thalassemia (β-TDT). We considered 68 β-TDT patients (11.98 ± 3.67 years, 51.5% females) consecutively enrolled in the Extension-Myocardial Iron Overload in Thalassemia network. Iron overload was quantified by T2* technique and biventricular function by cine images. Replacement myocardial fibrosis was evaluated by late gadolinium enhancement technique. Three groups of patients were identified: homozygous β+ (N = 19), compound heterozygous β0β+ (N = 24), and homozygous β0 (N = 25). The homozygous β0 group showed significantly lower global heart and pancreas T2* values than the homozygous β+ group. Compared to patients with homozygous β+ genotype, β0β+ as well as β0β0 patients were more likely to have pancreatic iron overload (odds ratio = 6.53 and 10.08, respectively). No difference was detected in biventricular function parameters and frequency of replacement fibrosis. No patient had cirrhosis/fibrosis, diabetes or heart failure, and the frequency of endocrinopathies was comparable among the groups. In pediatric β-TDT patients, there is an association between genotype and cardiac and pancreatic iron overload. The knowledge of patients’ genotype can be valuable in predicting some patients’ phenotypic features and in helping the clinical management of β-TDT patients.

## 1. Introduction

Beta-thalassemia is a genetic blood disease with a high incidence in the Mediterranean basin, Middle East, Indian subcontinent, Central Asia, and Far East [1]. However, due to migration, international job opportunities and intermarriages, thalassemia has become a global health problem. Beta-thalassemia is characterized by a wide spectrum of clinical manifestations and laboratory findings, and the disease phenotype largely depends on the underlying mutations of the β-globin gene [2]. More than 200 different mutations that affect the β-globin gene have been identified and their frequency varies significantly among countries or even in different regions of a single country [3]. Globally, in the Mediterranean Region the commonest mutations are CD39, IVS-1,110, IVS-1,1, IVS-1,6, IVS-2,745, IVS-2,1, and CD8, while CD41/42, IVS, 1-5, CD17, -28, COD8/9, IVS,2-654, and IVS-1,1 are the most prevalent mutations in the Asian Region [1]. These mutations cause a reduced (β+) or absent (β0) production of the β-globin chain, with relative excess of α-chains. The imbalance in the production of α- and β-globin chains results in ineffective erythropoiesis and peripheral red cell hemolysis, with consequent anemia [4]. However, the correlation between genotype and phenotype is complex, because other secondary/tertiary modifiers and environmental factors interact with the different allelic variants [2] and modulate the complex pathophysiology of β-thalassemia.

Based on the clinical severity and transfusion requirement, thalassemia can be classified into two main groups: transfusion-dependent thalassemia (TDT) and non-transfusion-dependent thalassemia (NTDT). TDT is the most severe clinical form of β-thalassemia, and its current treatment consists of regular transfusions (every 2–5 weeks) to maintain pre-transfusion hemoglobin levels above 9–10.5 g/dL [5]. Chronic transfusions are not risk-free and iron overload represents the main drawback. As the human body lacks an active mechanism to excrete excess iron, a progressive accumulation of body iron easily occurs as a result of long-term transfusions [6,7]. Iron overload is cytotoxic and induces organ damage and failure in the liver, heart, pancreas, thyroid, and the central nervous system [8]. The introduction of the chelation therapy has led to a decrease of the iron burden, significantly prolonging the survival of the patients [9]. Moreover, the deployment of the T2* magnetic resonance imaging (MRI) technique for the noninvasive quantification of organ-specific iron overload has offered the possibility to design tailor-made chelation therapies meeting the individual patient’s needs [10,11], further improving the prognosis [12,13]. However, hepatic, cardiovascular and endocrine complications still occur [13,14]. The underlying genetic defect in thalassemia is an important factor in the development of these complications because the homozygous β0 genotype state demands more red cells consumption and a greater rate of iron overloading [15,16].

A superior understanding of organ damage can be achieved by a comprehensive assessment of the children with TDT, representing the ideal population to study the initial stage of iron loading and the onset of its complications.

The aim of the present study was to evaluate the impact of the genotype on hepatic, pancreatic and myocardial iron content, and on hepatic, cardiac and endocrine complications in children with transfusion-dependent β-thalassemia.

## 2. Materials and Methods

### 2.1. Study Population

From the 1727 patients with hemoglobinopathies enrolled in the E-MIOT (Extension-Myocardial Iron Overload in Thalassemia) Network, we retrospectively selected 68 pediatric (age < 18 years) TDT patients who had undergone at least one MRI scan. We excluded the patients with Hb Lepore either in heterozygosis or in homozygosis and those with alfa abnormalities.

The E-MIOT Network is constituted by 66 thalassemia centres and 11 MRI sites where MR exams are performed using standardized and validated procedures [17,18]. All centers are linked by a web-based database configured to collect and share patients’ history, clinical and diagnostic data [19].

All patients were regularly transfused to maintain a pre-transfusion hemoglobin concentration above 9–10 g/dL. MRI scanning was performed within one week before regular scheduled blood transfusion. 

This study complied with the Declaration of Helsinki and was approved by the institutional ethic committees. Parents gave their informed consent for all patients.

### 2.2. MRI

MRI exams were performed on conventional clinical 1.5 T scanners of three main vendors (GE Healthcare, Philips Healthcare, Siemens Healthineers), equipped with a phased-array receiver surface coil. 

For iron overload assessment, a mid-transverse hepatic slice [20], five or more axial slices including the whole pancreas [21], and basal, medium and apical short-axis views of the left ventricle (LV) [22,23] were acquired by T2* gradient-echo multi-echo sequences. T2* image analysis was performed using a custom-written, previously validated software (HIPPO MIOT^®^, V2.0, Consiglio Nazionale delle Ricerche and Fondazione Toscana Gabriele Monasterio, Pisa, Italy) [24]. Hepatic T2* values were calculated in a circular region of interest (ROI) of standard dimension [25] and were converted into liver iron concentration (LIC) using the Wood’s calibration curve [26,27]. Three small ROIs were manually defined over pancreatic head, body and tail, taking care to avoid large blood vessels or ducts and areas involved in susceptibility artefacts from gastric or colic intraluminal gas [28]. Global pancreatic T2* value was calculated as the mean of T2* values from the three regions. The software provided the T2* value on each of the 16 segments of the LV, according to the standard AHA/ACC model [29]. The global heart T2* value was obtained by averaging all segmental T2* values. 

Steady-state free precession (SSFP) cine images were acquired in sequential 8-mm short-axis slices (gap 0 mm) from the atrio-ventricular ring to the apex to quantify biventricular function parameters in a standard way [30]. The inter-center variability had been previously reported [31]. Left and right atrial areas were measured from the 4-chamber view projection in the ventricular end-systolic phase.

To detect the presence of replacement myocardial fibrosis, late gadolinium enhancement (LGE) short-axis and vertical, horizontal, and oblique long-axis images were acquired 10–18 min after Gadobutrol (Gadovist^®^; Bayer; Berlin, Germany) intravenous administration at the standard dose of 0.2 mmol/kg using a fast gradient-echo inversion recovery sequence. The use of Gadobutrol has been demonstrated to be safe in patients with hemoglobinopathies [32]. LGE was considered present when visualized in two different views [33].

### 2.3. Biochemical Assays

All biochemical investigations were performed using commercially available kits at the laboratories of thalassemia centres where the patients were treated.

Genotyping was done using DNA sequencing techniques. Genomic DNA was extracted from peripheral blood leucocytes using the salting-out method [34]. All coding and noncoding regions of the β-globin gene were amplified by polymerase chain reaction (PCR) in different fragments. The PCR conditions were different, depending on the specific protocol adopted by the laboratory of thalassemia centre. Β-thalassemia mutations were identified by reverse hybridization assay (β-globin strip assay, Nuclear Laser, Vienna Lab, Austria).

The patients were monitored for glucose dysregulation according to the TIF guidelines [5], recommending an annual oral glucose tolerance test (OGTT) screening starting at age 10. Venous plasma glucose was measured fasting and 2 h after and oral glucose load (dose of 1.75 g/kg, with a maximum of 75 g).

### 2.4. Diagnostic Criteria

The compliance was collected by the investigators of each thalassemia center and, based on the correspondence between the patient’s actual dosing and the prescribed regimen, it was defined as excellent (> 80%), good (60–80%) or insufficient (<60%).

An MRI LIC ≥3 mg/g/dw was considered indicative of a significant iron load [35]. It was previously determined that 26 ms is the lowest threshold of a normal global pancreas T2* value [21]. A T2* measurement of 20 ms was taken as a “conservative” normal value for segmental and global values [24,36,37]. 

Liver fibrosis was diagnosed if the liver stiffness assessed by transient elastography was >7.0 kPa, while a liver stiffness >12.5 kPa was indicative of cirrhosis. 

Diabetes mellitus (DM) was defined by fasting plasma glucose ≥126 mg/dL or 2-h plasma glucose ≥200 mg/dL during an OGTT, or a random plasma glucose ≥200 mg/dL with classic symptoms of hyperglycemia or hyperglycemic crisis [38].

Hypogonadotropic hypogonadism was defined as luteinizing hormone (LH) and follicle stimulating hormone (FSH) levels below 2 IU/L, with an estradiol concentration of below 20 pg/mL in girls or a testosterone concentration of below 3 ng/mL in boys. Hypogonadism was detected in females by the absence of breast development and in males by the absence of testicular enlargement (<4 mL) by the age of 16 years [39].

Hypothyroidism was defined as a high serum TSH concentration with normal or reduced free thyroxine levels (primary form) or normal or low serum TSH concentration with reduced free thyroxine levels (central form) [40].

Hypoparathyroidism was defined as low serum calcium concentration, increased serum phosphate, low serum parathyroid hormone or, if normal, inappropriate for the calcium level [41].

Diagnosis of growth hormone (GH) deficiency required integration of growth criteria, medical history, laboratory tests (measurements of insulin-like growth factor 1 and insulin-like growth factor binding protein type 3 levels and provocative testing), and imaging studies [42]. 

Heart failure (HF) was identified based on symptoms and signs, according to the AHA/ACC guidelines [43]. Arrhythmias were diagnosed and classified according to the AHA/ACC guidelines [44].

### 2.5. Statistical Analysis

All data were analyzed using the SPSS v27.0 statistical package. 

Continuous variables were described as mean ± standard deviation (SD) and categorical variables were expressed as frequencies and percentages.

The normality of distribution of the parameters was assessed by using the Kolmogorov–Smirnov test.

For continuous values with normal distribution, comparisons among groups were made by one-way ANOVA. Kruskal–Wallis test was applied for continuous values with no normal distribution. The χ2 test was used for the comparison of non-continuous variables. Bonferroni post-hoc test was used for multiple comparisons between pairs of groups.

Correlation analysis was performed using Pearson’s test or Spearman’s test where appropriate. 

Odds ratios (OR) and 95% confidence intervals (CI) were calculated by using logistic regression.

In all tests, a two-tailed probability value of 0.05 was considered statistically significant

## 3. Results

### 3.1. Patients’ Characteristics

All patients were white and 35 (51.5%) were females. Mean age was 11.98 ± 3.67 years (range: 4–18 years). 

Thirty-three different genotypes were recorded and the commonest were homozygous CD39, CD39/IVS-1,110, and homozygous IVS-1,110 (Table 1). 

Each allele belonging to genotype was classified according to the corresponding phenotypic expression (β+ or β0) and patients were divided into three groups: homozygous β+ (N = 19; 27.9%), compound heterozygous β0β+ (N = 24; 35.3%), and homozygous β0 (N = 25; 36.8%).

### 3.2. Genotype and Clinical Correlates

The clinically relevant findings in the three groups are summarized in Table 2. Age, gender, frequency of splenectomy, mean pre-transfusion hemoglobin and serum ferritin levels were comparable among the groups. Age at the start of regular transfusions and chelation therapy tended to be lower in the homozygous β0 group, but the difference was not significant. 

The number of transfusional units in the 12 months before the MRI scan was available for 14 patients with homozygous β+ genotype (mean value: 31.79 ± 11.13), 18 patients with heterozygous β0β+ genotype (mean value: 37.18 ± 10.40), and 17 patients with homozygous β0 genotype (mean value: 38.89 ± 12.79) and, besides the trend, no association with the genotype was detected (*p* = 0.258).

No difference was found in the distribution of the different chelation regimens or in the compliance.

### 3.3. Genotype and MRI Findings

Mean MRI LIC was 7.19 ± 8.79 mg/g dw and hepatic iron overload was detected in 44 (64.7%) patients. In the 71 patients in whom pancreatic T2* images were available, mean global pancreas T2* value was 19.82 ± 11.54 ms, and pancreatic iron overload had an incidence of 73.1%. Mean global heart T2* value was 33.75 ± 10.49 ms. All seven (10.3%) patients with myocardial iron overload had both hepatic and pancreatic iron overload. 

Mean serum ferritin levels were directly correlated with MRI LIC values (R = 0.715; *p* < 0.0001) and inversely correlated with global pancreas T2* values (R = −0.442; *p* = 0.001) and global heart T2* values (R = −0.512; *p* < 0.0001). No association between iron overload in the different organs and age at start of regular transfusions or chelation was detected. 

MRI LIC values were inversely correlated with global pancreas T2* values (R = −0.561; *p* < 0.0001), as well as global heart T2* values (R = −0.595; *p* < 0.0001), and a positive correlation was detected between global pancreas and heart T2* values (R = 0.609; *p* < 0.0001). 

Table 3 summarizes the MRI findings in the three groups. MRI LIC values were comparable among the three groups.

Global pancreas T2* values were significantly lower in the homozygous β0 group than in the homozygous β+ group (*p* = 0.024) (Figure 1A). The percentage of patients with global pancreas T2* < 26 ms was significantly higher in both heterozygous β0β+ and homozygous β0 groups than in the homozygous β+ group (*p* = 0.018 and *p* = 0.003, respectively) (Figure 1B). The OR for abnormal global pancreas T2* values was 6.53 (1.59–26.79 95%CI; *p* = 0.009) for patients with the heterozygous β0β+ genotype and 10.08 (2.22–45.71%CI; *p* = 0.003) for patients with the homozygous β0 genotype versus patients with the homozygous β+ genotype.

The homozygous β0 group showed significantly lower global heart T2* values than the homozygous β+ group (*p* = 0.048) (Figure 2). In homozygous β0 patients, the frequency of myocardial iron overload was about three times higher than in patients with homozygous β+ genotype and two times higher than in patients with β0β+, but this difference was not significant. 

Biventricular function parameters were assessed in 63 patients, because for five patients a short MRI protocol was chosen to avoid sedation. There were not significant differences among groups in bi-atrial areas, biventricular volume indexes and ejection fractions, and LV mass index.

Eleven patients completed the MRI protocol with acquisition of the LGE images and only one of them showed replacement myocardial fibrosis.

### 3.4. Genotype and Complications

No patient had liver fibrosis or cirrhosis.

Diabetes mellitus was not diagnosed in any patient. Hypogonadism, hypothyroidism, and GH deficit developed in 5.9% of patients and prevalence of hypoparathyroidism was 4.4%. At least one endocrine complication was detected in 13 (19.1%) patients. Frequency of hypogonadism, hypothyroidism, hypoparathyroidism, GH deficit, and endocrinopathies globally considered was comparable among the three groups (Table 4).

No patient had heart failure. One patient had an ectopic atrial tachycardia which resulted in left ventricular dysfunction and required hospitalization and treatment with anti-arrhythmic drugs. This patient was a nine-year-old male with a homozygous β0 genotype and elevated iron levels (MRI LIC = 9.05 mg/g dw, global pancreas T2* = 4.20 ms, and global heart T2* = 2.40 ms).

## 4. Discussion

The knowledge of the molecular background of β-thalassemia can play a key role in the understanding of the factors affecting the diverse clinical manifestations. In the present study, we evaluated the impact of an underlying genetic defect on the development of disease complications in children with TDT. To the best of our knowledge, few studies, involving mainly patients from a limited area of Egypt, have explored this issue [14,16]. The children represent an ideal study population, due to the absence of the “aging” effect, with age “per se” being a powerful risk factor for the development of several complications [45,46].

We analyzed a representative sample of the Italian pediatric population with TDT, in which 33 different genotypes were identified. In line with previous studies on adult Italian patients [15,47], the most common mutations were CD39 and IVS-1,110. In the Egyptian studies, the commonest mutations were IVS-1,1, IVS-1,110, and IVS-1,6 [16].

Patients were divided into three groups according to their genotype: homozygous β0, with two severe mutations and consequently a high alpha-non alpha globin chain imbalance, heterozygous β0β+, characterized by a combination of mild/severe mutations, and homozygous β+, with two mild mutations and a lower imbalance between alpha and beta globin chains.

The three groups were homogeneous for age, sex, and hemato-chemical parameters. The transfusion demand tended to be lower in the homozygous β+ group than in the other two groups, but the difference was not statistically significant, likely because the datum was not available for all patients and the transfusion burden may change from center to center, depending also on red blood cell preparation and concentration. Moreover, it has been suggested that other factors (i.e., the spleen status) could contribute, more accurately than the genotype, to provide a basal evaluation of residual erythropoietic activity, and therefore, of the blood consumption [48]. Other studies found a significant association between genotype and frequency of blood transfusions, since in β0 homozygotes the complete absence of β-hemoglobin chains increases the degree of hemolysis and blood requests, also leading to a higher iron overload [16,49].

MRI LIC values tended to be higher in the homozygous β0 group, but the difference among the group was not significant. Conversely, global heart T2* values were significantly lower in the homozygous β° group than in the homozygous β+ group. Liver and heart have not only different iron uptake mechanisms, resulting in a “delay” in cardiac iron overload, but also different rates of iron clearance [50]. The capacity of the chelation therapy to remove iron from the liver in a faster way than from the heart is a potential explanation of our finding. In contrast with our study, Hassan et al. found that the homozygous β0 genotype was associated with significantly higher liver iron content compared to the heterozygous β0β+ and homozygous β+ genotypes [16]. These inconsistent results can be explained by differences in study populations. We considered only TDT patients, while 16.4% of their patients, all in the homozygous β+ group, had non-transfusion-dependent thalassemia (NTDT). Although iron overload can occur also in NTDT patients, due to the increased intestinal iron absorption, it occurs at a slower rate than in TDT patients [51]. Iron seems to be less adequately controlled in the Egyptian population in comparison to our patients, as highlighted by the higher mean serum ferritin levels (3385.8 vs. 1742.51 ng/mL) and mean MRI LIC values (17.4 vs. 7.19 mg/g dw). Moreover, the homozygous β0 group had a significantly worse compliance than the other two groups [16], and treatment efficacy and success are highly dependent on patient compliance [52].

To the best of our knowledge, this is the first study to show an association between genotype and levels of pancreatic iron overload. Likely, this link was not masked by the effects of the iron chelation therapy, since it seems extremely hard to remove iron from the pancreas [53]. We found out that, compared to patients with the homozygous β+ genotype, patients with the homozygous β0 genotype and patients with the β0β+ genotype had a risk ten and six times higher, respectively, to develop pancreatic iron overload. It could be hypothesized that the presence of at least one β+ allele, independently from the transfusional load, could be representative of a non-transfusion-dependent thalalassemia-like phenotype, characterized by the tendency to limit extrahepatic iron distribution [54,55,56,57]. Surprisingly, besides the young age, more than 80% of our patients with at least one β0 allele had pancreatic iron overload. Pancreatic iron is a powerful predictor for the alterations of glucose metabolism, although a latency time exists before pancreatic iron could cause impaired glucose tolerance and overt diabetes [53,58]. Moreover, pancreatic iron has a profound link with heart disease, being a good predictor for myocardial dysfunction in the absence of cardiac iron, for cardiac iron, for heart failure, and for arrhythmias [53,59,60,61]. Accordingly, it seems paramount to incorporate the pancreatic T2* assessment in the evaluation and monitoring of young children with TDT, especially in the presence of a β0 allele. Thalassemic children were shown to be prescribed a lower mean dose of chelating drug and a delayed dose increase in comparison with adult patients [62], but in the presence of pancreatic iron overload it would be prudent to modify or intensify the iron chelation therapy to prospectively ward off both alterations of glucose metabolism and cardiac iron accumulation.

We failed to detect a correlation between genotype and biventricular volumes and ejection fractions and LV mass index, likely because all patients had normal or near-normal values of these parameters.

In our study population, no patient had liver fibrosis or cirrhosis and only one patient had hepatitis C virus (HCV) infection, confirming that in Italy the implemented measures to improve blood transfusion screening have significantly reduced the risk of getting an infection. Indeed, in an Italian study involving 1079 TDT patients with a mean age of 37.79 ± 10.11 years, only 37.1% had never contracted the HCV infection [53]. In the same study, the prevalence of diabetes mellitus was 17.5%. Conversely, none of our pediatric patient showed diabetes mellitus. In thalassemia, diabetes is a late complication, and its prevalence is significantly associated with the age of the patient [63,64,65].

The prevalence of the other endocrine complications in our pediatric population was hypogonadism, hypothyroidism and GH deficit—5.9%, and hypoparathyroidism—4.4%. This prevalence was consistent with some studies [66,67], while other studies reported a significantly higher incidence of endocrinopathies [14,16]. These discrepancies could be attributed to differences in age distribution and therapeutic management, and to genetic, geographical, cultural, and economic factors. In pediatric as well as in adult thalassemia patients, the β0β0 genotype was previously demonstrated to be associated with a higher rate of endocrinopathies [14,16,49,63,68], likely due to a greater rate of iron loading through higher red cell consumption and a higher vulnerability to free radical damage. It is highly probably that, in the present study, this association is hidden by the low number of patients with endocrinopathies.

We were not able to explore the association between genotype and cardiac complications, since no patient had heart failure and only a single patient had arrythmias. Of note, this patient had a homozygous β0 genotype, which, in a prospective multicenter study, emerged as a risk factor for the development of cardiac arrhythmias and complications globally considered [69].

The main limitation of this study is the small sample size. Moreover, the previous complete transfusion history and chelation therapy, as potential determinants of the described pattern of iron organ distribution, were not available.

## 5. Conclusions

In pediatric TDT patients there is an association between genotype and cardiac and pancreatic iron overload. Patients with homozygous β+ genotype had significantly lower myocardial iron levels than β0β0 patients and significantly lower pancreatic iron levels than β0β+ and β0β0 patients. So, the knowledge of the genotype can be valuable in predicting some patients’ phenotypic features and in helping the clinical and instrumental management of TDT patients.

## Figures and Tables

**Figure 1 jpm-12-00400-f001:**
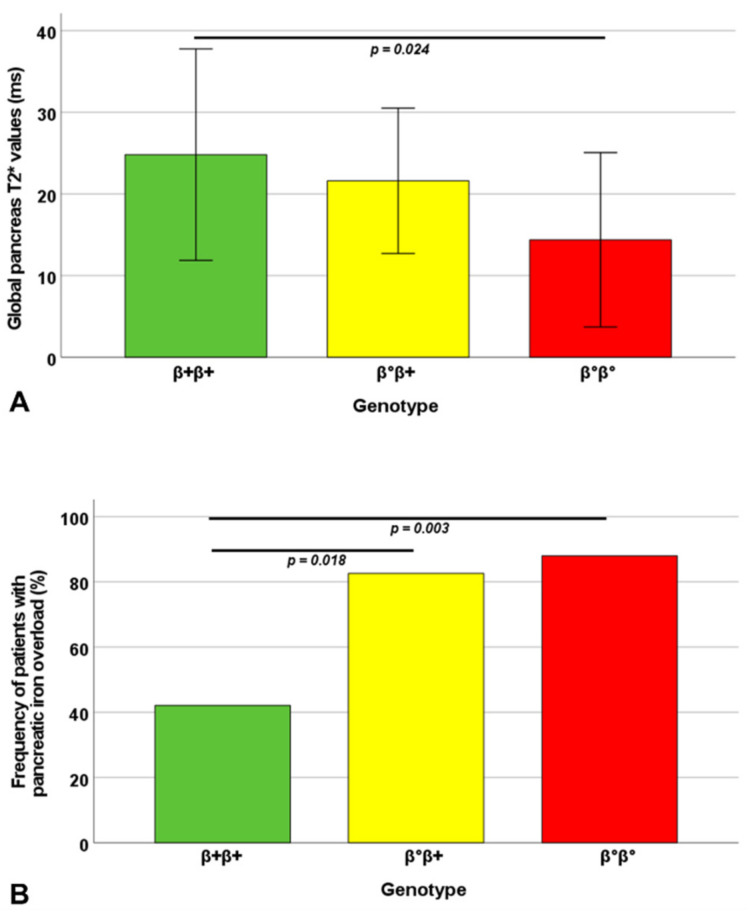
(**A**) Mean global pancreas T2* values in the three groups identified based on the β-globin gene phenotypic expression. (**B**) Frequency of patients with global pancreas T2* < 26 ms in the three phenotypic groups. The *p*-value for each significant pairwise comparison is indicated.

**Figure 2 jpm-12-00400-f002:**
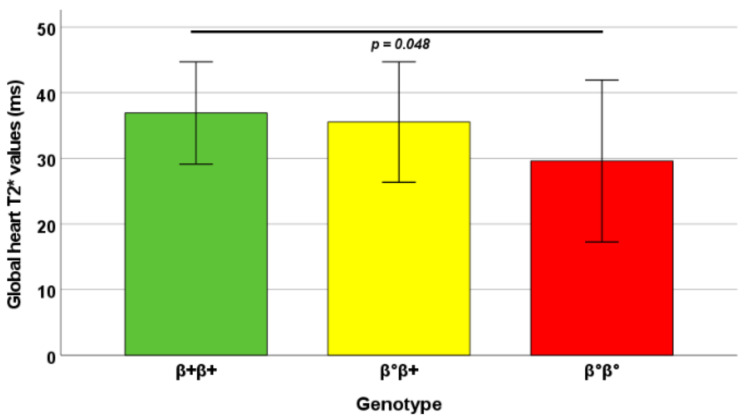
Mean global heart T2* values in the three groups identified based on the β-globin gene phenotypic expression. The *p*-value for each significant pairwise comparison is indicated.

**Table 1 jpm-12-00400-t001:** Frequency of different genotypes (based on type of mutation) in our pediatric TDT patients.

Genotype	HGVS Nomenclature	Type	Cases (N)	Frequency (%)
CD39/CD39	HBB:c.118C > T/HBB:c.118C > T	β0 β0	11	16.2
CD39/IVS-1,110	HBB:c.118C > T/HBB:c.93-21G > A	β0 β+	7	10.3
IVS-1,110/IVS-1,110	HBB:c.93-21G > A/HBB:c.93-21G > A	β+ β+	6	8.8
IVS-1,110/IVS-2,745	HBB:c.93-21G > A/HBB:c.316-106C > G	β+ β+	4	5.9
CD5/CD5	HBB:c.17_18delCT/HBB:c.17_18delCT	β0 β0	3	4.4
CD39/IVS-1,1	HBB:c.118C > T/HBB:c.92 + 1G > A	β0 β0	3	4.4
IVS-1,6/IVS-1,110	HBB:c.92 + 6T > C/HBB:c.93-21G > A	β+ β+	3	4.4
CD39/IVS-1,6	HBB:c.118C > T/HBB:c.92 + 6T > C	β0 β+	3	4.4
CD6/-87	HBB:c.20delA/HBB:c.-137C > G	β0 β+	2	2.9
CD39/IVS-2,1	HBB:c.118C > T/HBB:c.315 + 1G > A	β0 β0	2	2.9
IVS-1,6/IVS-1,6	HBB:c.92 + 6T > C/HBB:c.92 + 6T > C	β+ β+	2	2.9
IVS-2,1/IVS-1,110	HBB:c.315 + 1G > A/HBB:c.93-21G > A	β0 β+	2	2.9
Others		β+ β+ β0 β+ β0 β0	4 10 6	5.9 14.8 8.8

HGVS = Human Genome Variation Society; N = number.

**Table 2 jpm-12-00400-t002:** Demographic, transfusion, chelation and clinical characteristics in the three groups identified on the basis of the β-globin gene phenotypic expression.

	β+β+ (N = 19)	β0β+ (N = 24)	β0β0 (N = 25)	*p*
Age (years)	12.45 ± 3.69	13.13 ± 3.30	10.53 ± 3.65	0.071
Males/Females	10/9	13/11	10/15	0.559
Age at start of regular transfusion (months)	20.46 ± 29.78	19.74 ± 17.69	12.44 ± 11.83	0.339
Chelation starting age (years)	4.92 ± 5.62	3.41 ± 2.24	2.94 ± 0.93	0.958
Splenectomy, N (%)	1 (5.3)	3 (12.5)	3 (12.0)	0.696
Positive HCV RNA (%)	0 (0.0)	0 (0.0)	1 (4.0)	0.418
Chelation therapy, N (%)				0.299
DFO	2 (10.5)	0 (0.0)	0 (0.0)	
DFP	4 (21.1)	3 (12.5)	2 (8.0)	
DFX	12 (63.2)	18 (75.0)	21 (84.0)	
Combined DFO + DFP	0 (0.0)	1 (4.2)	2 (8.0)	
Sequential DFO/DFP	0 (0.0)	1 (4.2)	0 (0.0)	
Combined DFP + DFX	1 (5.3)	1 (4.2)	0 (0.0)	
Compliance, N (%)				0.905
optimal	10 (52.6)	10 (41.7)	11 (44.0)	
good	8 (42.1)	12 (50.0)	11 (44.0)	
insufficient	1 (5.3)	2 (8.3)	3 (12.0)	
Pre-transfusion hemoglobin (g/dL)	9.93 ± 0.46	9.52 ± 0.51	9.59 ± 0.58	0.072
Ferritin levels (ng/L)	1684.14 ± 1276.53	1655.56 ± 1518.69	1886.17 ± 1805.39	0.901

N = number; HCV = hepatitis C virus; RNA = ribonucleic acid; DFO = desferrioxamine; DFP = deferiprone; DFX = deferasirox.

**Table 3 jpm-12-00400-t003:** MRI findings in the three groups identified on the basis of the β-globin gene phenotypic expression.

	β+β+ (N = 19)	β0β+ (N = 24)	β0β0 (N = 25)	*p*
MRI LIC (mg/g dw)	5.39 ± 5.59	6.67 ± 10.29	9.07 ± 9.19	0.140
MRI LIC > 3 mg/g dw, N (%)	12 (63.2)	13 (54.2)	18 (72.0)	0.433
Global pancreas T2* (ms)	24.80 ± 12.95	21.60 ± 8.89	14.38 ± 10.68	0.006
Global pancreas T2* < 26 ms, N (%)	8 (42.1)	19/23 (82.6)	22 (88.0)	0.001
Global heart T2* (ms)	36.93 ± 7.78	35.53 ± 9.17	29.61 ± 12.35	0.042
Global heart T2* < 20 ms, N (%)	1 (5.3)	2 (8.3)	4 (16.0)	0.472
Left atrial area (cm^2^/m^2^)	12.54 ± 1.21	12.65 ± 1.96	13.49 ± 3.97	0.881
Right atrial area (cm^2^/m^2^)	11.97 ± 2.51	11.53 ± 1.29	12.29 ± 3.98	0.964
LV EDVI (mL/m^2^)	80.84 ± 12.82	80.65 ± 12.89	79.87 ± 15.02	0.874
LV ESVI (mL/m^2^)	34.26 ± 11.85	31.57 ± 6.85	29.23 ± 7.69	0.238
LV SVI (mL/m^2^)	49.05 ± 8.92	50.80 ± 9.14	49.73 ± 8.22	0.899
LV mass index (g/m^2^)	51.02 ± 13.66	52.00 ± 12.95	51.23 ± 10.38	0.963
LV EF (%)	60.62 ± 6.69	61.87 ± 4.39	62.99 ± 4.31	0.334
LV cardiac index (L/min/m^2^)	4.35 ± 0.81	4.02 ± 1.04	4.24 ± 0.75	0.484
RV EDVI (mL/m^2^)	82.41 ± 12.52	80.48 ± 15.49	76.36 ± 14.23	0.390
RV ESVI (mL/m^2^)	32.98 ± 7.99	30.57 ± 6.78	27.64 ± 6.28	0.062
RV SVI (mL/m^2^)	49.80 ± 9.89	49.78 ± 10.67	49.72 ± 8.91	0.950
RV EF (%)	60.27 ± 7.83	61.35 ± 5.29	63.57 ± 3.50	0.181
Replacement myocardial fibrosis, N (%)	0/1 (0.0)	1/5 (20.0)	0/5 (16.7)	0.517

N = number; MRI = magnetic resonance imaging; LIC = liver iron concentration; LV = left ventricular; EDVI = end-diastolic volume index; ESVI = end-systolic volume index; SVI = stroke volume index; LV = ejection fraction; RV = right ventricular.

**Table 4 jpm-12-00400-t004:** Hepatic, endocrine and cardiac complications in the three groups identified on the basis of the β-globin gene phenotypic expression.

	β+β+ (N = 19)	β0β+ (N = 24)	β0β0 (N = 25)	*p*
Liver fibrosis or cirrhosis, N (%)	0 (0.0)	0 (0.0)	0 (0.0)	–
Diabetes mellitus, N (%)	0 (0.0)	0 (0.0)	0 (0.0)	–
Hypogonadism, N (%)	0 (0.0)	2 (8.3)	2 (8.0)	0.438
Hypothyroidism, N (%)	1 (5.3)	2 (8.3)	1 (4.0)	0.805
Hypoparathyroidism, N (%)	0 (0.0)	2 (8.3)	1 (4.0)	0.414
GH deficit, N (%)	1 (5.3)	1 (4.2)	2 (8.0)	0.842
At least one endocrinopathy, N (%)	2 (10.5)	6 (25.0)	5 (20.0)	0.483
Heart failure, N (%)	0 (0.0)	0 (0.0)	0 (0.0)	–
Arrhythmias, N (%)	0 (0.0)	0 (0.0)	1 (4.0)	0.418

N = number; GH = growth hormone.

## Data Availability

The data presented in this study are available on request from the corresponding author. The data are not publicly available due to privacy.

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
