# Peer review of "Link between Genotype and Multi-Organ Iron and Complications in Children with Transfusion-Dependent Thalassemia"

_jpm, 2022, doi:10.3390/jpm12030400_

Round 1

Reviewer 1 Report

In this paper, Meloni et al. aimed to evaluate the link between genotype and hepatic, pancreatic and cardiac complications notably related to iron in transfusion-dependent thalassemic pediatric patients.

The introduction is very well written. The background is well explained, and the references are well chosen. The aim of the study is clearly stated. Nevertheless, a short description of the most frequent mutations should be added: CD39; IVS1-1, CD5…

The result section is clear too. Do the authors have any feedback on the spleen status such as circulating Howell Jolly bodies and spleen scintigraphy.

As perspective, do the authors plan to perform a longitudinal study of the patients? It would great to evaluate the evolution of the cardiac function.

As some data are missing such as the transfusion history and the chelation therapy, it would be insightful to perform a prospective study.

Author Response

We would like to thank the Editor and the Reviewer for their encouraging feedback and constructive critique and for the effort regarding this manuscript. We have addressed each of the raised concerns, which have substantially improved the manuscript.

In this paper, Meloni et al. aimed to evaluate the link between genotype and hepatic, pancreatic and cardiac complications notably related to iron in transfusion-dependent thalassemic pediatric patients.

The introduction is very well written. The background is well explained, and the references are well chosen. The aim of the study is clearly stated. Nevertheless, a short description of the most frequent mutations should be added: CD39; IVS1-1, CD5…

A: We thank the Reviewer for this comment. As requested, the following sentence has been added in the Introduction. Globally, in the Mediterranean Region the commonest mutations are CD39, IVS-1,110, IVS-1,1, IVS-1,6, IVS-2,745, IVS-2,1, and CD8 while CD41/42, IVS,1-5, CD17, -28, COD8/9, IVS,2-654, and IVS-1,1 are the most prevalent mutations in the Asian Region [1].

The result section is clear too. Do the authors have any feedback on the spleen status such as circulating Howell Jolly bodies and spleen scintigraphy.

A: Unfortunately, we don’t’ have data on the spleen status.

As perspective, do the authors plan to perform a longitudinal study of the patients? It would great to evaluate the evolution of the cardiac function.

A: We will perform a longitudinal study.

As some data are missing such as the transfusion history and the chelation therapy, it would be insightful to perform a prospective study.

A: We strongly agree with this comment and we will surely perform a prospective study.

Reviewer 2 Report

This research aims to demonstrate the association between genotype of beta-thalassemia and iron accumulation in vital organs including liver, heart and pancreas with MRI T2* technique. The key finding of this research is the association between ß0ß0 genotype with pancreatic iron overload. Although the author failed to demonstrate the organ dysfunctions in the different genotypes, it was not surprised because the development of those may observe after the second decade of life. Generally, the manuscript is well written, good experimental design.

It is known that genotype ß0ß0 leads to sever anemia and the patients may need blood transfusion earlier than others. Therefore, the heavier iron overload in several organs might not directly relate to the genotype per se. The organ iron distribution may different in the individual depending on several factors; however, it is difficult to conclude that the genotype can predict the patients’ phenotype in this case.

The correlation analysis of MRI T2* among the studied organs and  correlation analysis with iron status (ferritin), frequency and amount of blood transfusion may help or support this conclusion.

Minor:

Some mistypes, for example table 3, column 1, row 2 and 3 with same description.

Author Response

We would like to thank the Editor and the Reviewer for their encouraging feedback and constructive critique and for the effort regarding this manuscript. We have addressed each of the raised concerns, which have substantially improved the manuscript.

This research aims to demonstrate the association between genotype of beta-thalassemia and iron accumulation in vital organs including liver, heart and pancreas with MRI T2* technique. The key finding of this research is the association between ß0ß0 genotype with pancreatic iron overload. Although the author failed to demonstrate the organ dysfunctions in the different genotypes, it was not surprised because the development of those may observe after the second decade of life. Generally, the manuscript is well written, good experimental design.

A: We thank the Reviewer for this comment.

It is known that genotype ß0ß0 leads to sever anemia and the patients may need blood transfusion earlier than others. Therefore, the heavier iron overload in several organs might not directly relate to the genotype per se. The organ iron distribution may different in the individual depending on several factors; however, it is difficult to conclude that the genotype can predict the patients’ phenotype in this case.

The correlation analysis of MRI T2* among the studied organs and  correlation analysis with iron status (ferritin), frequency and amount of blood transfusion may help or support this conclusion.

A: We agree that several factors, as the chelation therapy, can influence the organ iron distribution. In fact, we did not find any association between genotype and hepatic iron overload.

The following sentences have been added in the Results section. “Mean serum ferritin levels were directly correlated with MRI LIC values (R=0.715; P<0.0001) and inversely correlated with global pancreas T2* values (R=-0.442; P=0.001) and global heart T2* values (R=-0.512; P<0.0001). No association between iron overload in the different organs and age at start of regular transfusions or chelation was detected.”

Unfortunately, as stated in the Discussion, we don’t have data on the previous complete transfusion history

Minor:

Some mistypes, for example table 3, column 1, row 2 and 3 with same description.

A: We have now deleted the wrong row.

Reviewer 3 Report

no comments

Author Response

Thank you for appreciating our work.